# Colour Fastness to Various Agents and Dynamic Mechanical Characteristics of Biocomposite Filaments and 3D Printed Samples

**DOI:** 10.3390/polym13213738

**Published:** 2021-10-29

**Authors:** Deja Muck, Helena Gabrijelčič Tomc, Urška Stanković Elesini, Maruša Ropret, Mirjam Leskovšek

**Affiliations:** Faculty of Natural Sciences and Engineering, University of Ljubljana, Aškerčeva Cesta 12, 1000 Ljubljana, Slovenia; helena.gabrijelcic@ntf.uni-lj.si (H.G.T.); urska.stankovic@ntf.uni-lj.si (U.S.E.); marusa.ropret@gmail.com (M.R.); mirjam.leskovsek@ntf.uni-lj.si (M.L.)

**Keywords:** biocomposites, PLA, wood fibres, hemp fibres, 3D printing, colour fastness, dynamic mechanical analysis

## Abstract

The aim of the study was to analyse the colour fastness of 3D printed samples that could be used as decorative or household items. Such items are often fabricated with 3D printing. The colour of filaments affects not only the mechanical properties, but also the appearance and user satisfaction. Samples of biocomposite filaments (PLA and PLA with added wood and hemp fibres) were used. First, the morphological properties of the filaments and 3D printed samples were analysed and then, the colour fastness against different agents was tested (water, oil, detergent, light and elevated temperature). Finally, the dynamic mechanical properties of the filaments and 3D printed samples were determined. The differences in the morphology of the filaments and 3D printed samples were identified with SEM analysis. The most obvious differences were observed in the samples with wood fibres. All printed samples showed good resistance to water and detergents, but poorer resistance to oil. The sample printed with filaments with added wood fibres showed the lowest colour fastness against light and elevated temperatures. Compared to the filaments, the glass transition of the printed samples increased, while their stiffness decreased significantly. The lowest elasticity was observed in the samples with wood fibres. The filaments to which hemp fibres were added showed the reinforcement effect. Without the influence on their elasticity, the printed samples can be safely used between 60 and 65 °C.

## 1. Introduction

In recent decades, interest in renewable biopolymers has increased in both industry and academia. Natural fibres offer an environmentally friendly alternative to conventional fibre reinforcement fibres (e.g., glass, Kevlar, carbon) [1].

Increasingly high raw material prices also increase the need to use natural, renewable materials in the development and production of polymer composites [2]. The distinctive features of biocomposites are their ecological properties, which make them environmentally friendly, fully degradable and sustainable [1]. Therefore, polymers from renewable resources are increasingly used as matrices for biocomposite materials. The most important in terms of quantity are starch-based polymers, i.e., PLA and PHA [2]. PLA is a biodegradable thermoplastic polymer with excellent mechanical properties. Under the right conditions, PLA decomposes into carbon dioxide, water and methane. PLA has long been the subject of biomedical research and in recent years, it has also been used as a matrix for composite materials [1].

A review of the literature shows that the treatment of colour and colour fastness in biocomposites has not been a research priority in recent years. In addition, research addressing the impact of PLA filament colours on printed product properties is very scarce. In products such as household appliances, jewellery, clothing accessories and toys [3,4], where there is a great potential for the application of biocomposites, and colour and its durability are important properties that affect not only the appearance, but also the satisfaction and quality of the experience [5]. The relationships between colour values and colour differences at different stages of the product use cycle, and the mechanical and chemical properties are important as well. Particularly in the case of products for domestic use, toys and accessories, the material comes into contact not only with light, moisture, dust and dirt, but also with water, cleaning agents (e.g., detergents), oils, creams and saliva.

Natural fibres are obtained from renewable resources can be recycled, have lower density, are cheaper and healthier and therefore also contribute to better working conditions. Despite the above advantages, natural fibres also present some obstacles when used in polymer composites, i.e., high sensitivity to moisture and water absorption, high temperature sensitivity, and poor adhesion between the hydrophilic fibres and the hydrophobic polymer matrix. Attempts have been made to overcome these obstacles using various fibre processing methods, mainly chemical. Among the most commonly used methods are silane treatment, esterification and the use of compatibilizers [6].

More and more biocomposite materials are also used in 3D printing. In the most widespread technology of 3D printing, i.e., fused deposition modelling (FDM), we can find an increasing number of thermoplastic biocomposite filaments on the market, which are used for printing with the mentioned technology.

Several studies in the literature describe the use of different forms of cellulose (cellulose microfibers, microcrystalline cellulose, cellulose nanocrystals, etc.) in combination with PLA as a matrix for the production of PLA micro- and nanocomposites. Cellulose biocomposites are popular due to their structural reinforcement, lightness, biodegradability and non-toxicity. However, the use of cellulose to reinforce PLA polymers is still a relatively new area of research. A major problem is the hydrophilicity of cellulose fibres, which, as mentioned earlier, leads to poorer bonding with the polymer matrix and thus poorer mechanical properties [7]. Moisture absorption increases with fibre content and temperature and is also affected by the processing of fibres and their distribution. Moisture absorption reduces the mechanical stability of biocomposites [8]. According to the literature, the most important factors affecting the properties of composites with natural fibres are the selection of fibres (type, processing, fibre content, annual time of extraction, etc.), the selection of the matrix, the intermolecular bonds, the distribution of fibres in the matrix, the orientation of fibres, the manufacturing process of the biocomposites and porosity [1,9,10].

Hemp fibres are among the cheapest and most affordable fibres, which is why they attract the most interest in the automotive industry as a polymer reinforcement. Polypropylene (PP) is usually used as the matrix since it has good properties, e.g., in terms of density [6]. In tests with PP composites and hemp fibres, the bonds between the polymer and the fibres could cause problems in the structure; therefore, a number of selected compatibilizers have to be added to the matrix. Compared to the composites without additives, the former were found to be more thermally stable and all composites also showed higher tensile strength compared to pure PP [2]. Additionally, improvement in the tensile strength of PLA is possible with the addition of hemp [10], whereas the addition of a plasticiser has been suggested to decrease pore size and improve surface finish, layer adhesion and performance of the composite [11]. Studies on PLA matrices reinforced with hemp fibres have shown that aligned fibres absorb less moisture due to their lower porosity. Moreover, an alkaline treatment has also been shown to reduce moisture absorption [8].

Pine, spruce and fir fibres are most commonly used for wood composites. Wood dust or very short fibres are used in composites, which are usually generated as residues in wood processing plants [12].

Filaments with added wood fibres are usually PLA-based composites. The percentage of wood fibres is usually around 30%, although the exact amount varies depending on the filament manufacturer. These wood particles give the finished printed products the appearance of wood. Filaments with added wood fibres have the property that they change colour at different extrusion die temperatures, i.e., they become darker at higher temperatures. These materials are printed in the same way as ordinary PLA, at the same temperatures, and an adhesive must also be applied to the table to improve the adhesion of lower layers. The addition of wood particles brings its own challenges and also requires some finishing (e.g., sanding) to give the materials a natural wood appearance [13,14].

Printing with the FDM technology and wood fibre biocomposites is very important for the development of technology and materials. A study was conducted to investigate the effect of printing parameters on wood fibre biocomposite filaments. It was found that the filament has a very porous structure and the printed objects show a similar porous structure. The porosity of printed objects could be mitigated by printing at a 90-degree angle. The porosity is also affected by the width of the test object. The orientation of wood fibres corresponds to the orientation of the filament, and the weakest link in the objects printed with the wood fibre filament is the bonds between the layers of the printed object. Higher porosity and poorer cohesive forces result in poorer mechanical properties and higher moisture absorption of printed objects. The FDM technology presents some challenges to the usability of high-quality printed objects, as the objects have low mechanical resistance and are sensitive to moisture. However, moisture sensitivity can also be considered an advantage as the field of the so-called 4D printing evolves. 4D printing is the ability of 3D printed objects to be “activated” by exposure to an external factor. This results in passive hygromorphic products, often inspired by the objects in nature, the shape of which is altered by moisture [15].

The authors Wittbrodt and Pearce [16] investigated the effects of colour of commercial PLA filaments (3 mm) on crystallinity, tensile strength and microstructure. The study was conducted using PLA filaments in natural, white, black, silver and blue colours from LutzBot printed on an open-source printer with 100% fill, alternating fill pattern and temperatures between 190 and 215 °C. The results of the study showed that white PLA had the highest crystallinity, followed by blue, grey and black. Natural PLA exhibited the lowest crystallinity. The regression line analysis of stress and strain showed that there was a significant correlation between the tensile strength and strain for different colours. The results also indicate that the temperature should be optimised depending on the filament colour to obtain the optimal mechanical properties of the prints.

The actual application value of colours is also proven by the research of the authors Pandžić et al. [17]. They studied the influence of colours of PLA filaments on the mechanical properties. Specimens were made with ISO 527-2 and printed on an Ultimaker 2+ desktop printer with the identical conditions for all colours. Fourteen different PLA colours were used (green, blue, white, orange, pink, red, gold, grey, silver, brown, purple, black and yellow). The results showed that PLA colour had an effect on the Young’s modulus (it varied by up to 18%, with red being the highest and pink the lowest), yield strength (it varied by up to 36%, with red being the highest and pink the lowest), tensile strength (it varied by up to 31%, with red being the highest and pink the lowest), toughness (it varied by over 300% depending on the colour, with pink being the highest and blue the lowest) and elongation (it varied by over 400% depending on the colour, with pink being the highest and blue the lowest).

Hanon et al. [18] investigated the accuracy of 3D printing depending on the colours of PLA filaments (1.75 mm, manufacturer eSUN). They chose three colours (white, grey and black) of PLA filaments that were printed on an FDM printer (Wanhao Duplicator 6). After the printing, the dimensional accuracy for cylindrical and dog-bone samples was tested. The results revealed that the colour of filaments had an influence on weight. The white specimen was the heaviest and the black the lightest. The greatest differences in dimensional accuracy with respect to filament colour were seen at a 45° pressure angle, while lower values were measured at horizontal and vertical orientations. White filaments generally had the lowest dimensional accuracy, followed by grey, while black filaments had the best accuracy.

No research was found in the field of 3D printing that addressed colour values in relation to the influence of factors such as water, oil, detergent; hence, we also examined related fields to get insight about the possible methodology.

Tor-Świątek and Garbacz [19] examined colorimetrically PLA composites with linen fibres (0–5% of linen addition, in steps of 1%) that were exposed to abiotic degradation. The samples were prepared with injection moulding. Besides SEM morphological analysis, the CIELAB model for the determination of lightness, colour saturation, chromatic colours and colour differences was used. After the thermal degradation, colorimetric images and spectral profiles were examined. The results of the influence of the percentage of linen content on the morphology and mechanical properties were presented with colour differences, values L*, a* and b*, and colour saturation.

The aim of this research was to determine the colour fastness of 3D printed samples that can be used as decorative and/or useful household items, e.g., household utensils, jewellery, clothing accessories, unique art products and toys. 

Nowadays, with increasing environmental awareness, more and more decorative and useful items are being produced with 3D printing using biocomposite materials. With the technology of extruding biocomposite filaments containing various additives, beautiful objects can be made; however, their durability remains questionable.

## 2. Materials and Methods

### 2.1. Materials

The research was conducted using three commercial biocomposite filaments and three-dimensional (3D) samples printed from commercial biocomposite filaments.

Commercial biocomposite filaments are marked with the suffix “_f” in their names and 3D samples with the suffix “_3D”.

#### 2.1.1. Commercial Biocomposite Filaments 

In the research, filaments (_f) with a poly(lactic acid) (PLA) matrix were used:-PLA_f: the neat sample consisted of pure poly(lactic acid) supplied by Plastika Trček (Slovenia) with a density of 1.26 g/cm^3^ and thickness of 2.85 mm, according to the technical data.-PLA-Woodfill_f: the biocomposite sample was supplied by ColourFabb (Netherlands). It consisted of poly(lactic acid) as a matrix with 30% wood fibres. According to the technical data, the density of the filament was 1.15 g/cm^3^ and the thickness was 2.85 ± 0.05 mm.-PLA-Entwined_f: the biocomposite sample was supplied by 3D Fuel (Ireland). It consisted of poly(lactic acid) as a matrix with <10% of hemp by-products, including hurd and fibre, with a thickness of 1.75 mm, according to the technical data.

### 2.1.2. 3D Printed Samples

The design for test samples was according to the recommendation of different standards modelled in Blender, version 2.73 (Blender Foundation, Amsterdam, The Netherlands). All samples were then printed with the fused deposition modelling (FDM) technology on a ZMorph 2.0 SX 3D printer (ZMorph, Wrocław, Poland) and the print files were prepared in Voxelizer (ZMorph, Wrocław, Poland). The samples were printed under the same conditions: extrusion nozzle temperature 195 °C and table temperature 80 °C. The samples were printed with a line pattern and full fill density (100%) at 200 µm layer thickness. As mentioned above, the 3D printed samples were named PLA_3D, PLA-Woodfill_3D and PLA-Entwined_3D.

### 2.2. Methods

The ***morphological properties*** of the filaments and 3D printed samples were observed using a scanning electron microscope, JSM-6060LV (Jeol, Japan) (SEM). For observation at different magnifications, the samples were covered with an ultrathin layer of gold (with high vacuum evaporation). 

***Influence of printing process on colour fastness***. The colour values were measured first on the filaments and then on the samples printed with these filaments. The colour differences were calculated based on these measurements. The measurements were made on ten samples for both the filaments and the printed samples, and the average values (L*, a* and b*) were calculated. L* indicates lightness, a* is the red/green coordinate, and b* is the yellow/blue coordinate. Based on the average values, the colour differences (ΔE*_a,b_) using CIE L*a*b* coordinates between the 3D printed samples and the filaments were calculated. An i1Pro 2 spectrophotometer (X-rite, Grand Rapids, MI, USA) was used to measure the colour values. The colour difference was calculated according to Equation (1).
(1)$$\Delta E_{a,b}^{*}=\sqrt{\Delta L^{*2}+\Delta a^{*2}+\Delta b^{*2}}$$

The ***colour fastness*** of the samples ***in water, oil and detergent*** was tested in accordance with the standard SIST ISO 2836: 2004 Graphic technology—Prints and printing inks—Assessment of resistance of prints to various agents. Tap water, vegetable sunflower oil and ordinary kitchen dishwashing detergent at a concentration of 1% were used for the test. A glass plate (60 × 90 × 2 mm) was covered with two filter papers (60 × 90 mm) soaked separately with water/oil/detergent. A 3D printed sample was placed on top of the filter papers, followed by two soaked filter papers. Finally, everything was covered with another glass tile. The prepared composition was weighted with a kilogram weight. The colour fastness test took 24 h for water, 48 h for oil and 3 h for detergent.

From each filament, five 3D printed samples with the dimensions of 20 × 50 × 3 mm were printed. Three samples were exposed to water/oil/detergent and two served as reference samples. Weight and spectrophotometric values were measured before and after the testing. The measurements were performed using an i1Pro 2 spectrophotometer (X-rite, Grand Rapids, MI, USA) under D50 light and 2° observer. Three measurements were taken for each sample and the average values were calculated. The colour difference was calculated according to Equation (1).

***The light fastness*** of the 3D printed samples was tested according to the standard SIST ISO 12040. A Xenotest Alpha (Atlas, Mount Prospect, IL, USA) was used. Two samples (45 × 15 × 2 mm) were 3D printed from single filaments. The samples were illuminated at the wavelength of 300–800 nm for 72 h. The colour values L*, a* and b* of the illuminated and non-illuminated part of the samples were then measured using an i1Pro 2 spectrophotometer (X-rite, Grand Rapids, MI, USA). The colour differences were again calculated using Equation (1).

***Temperature resistance*****.** The samples were printed in the form of a standardised 75 mm long and 10 mm wide tube with a narrow part of 5 mm in width, according to ISO 527-2. Twenty samples were printed with each filament. Five samples were not exposed and served as references. Five samples were exposed to higher temperatures (80 °C, 110 °C and 130 °C) for 1 week. The samples were evaluated spectrophotometrically before and after the exposure to higher temperatures.

***Analysis of mechanical properties***. The mechanical properties of the samples were tested according to the standard ISO 13934-1:2013. For this purpose, the same samples were used as for the analysis of temperature resistance, where colour differences were evaluated. An Instron instrument was used to measure the breaking force, elongation, extension and Young’s modulus at a restraint length of 40 mm and speed of 0.1 mm/s. The width of the samples was set to 5 mm and the thickness to 2.21 mm. For the test, the samples were clamped and stretched using the Instron software, and the specimens disintegrated after a certain time.

The testing included the ***dynamic mechanical analysis (DMA)*** of the filaments and 3D printed samples to measure the mechanical and viscoelastic properties of the materials. The measurements were performed using a Q800 DMA analyser (TA Instruments, New Castle, DE, USA) in a single cantilever bending mode on the samples with a length of 17.5 mm, at 10 Hz oscillation frequency, 10 µm oscillation amplitude and the temperature ramp of 3 °C/min in the range from 0 to 120 °C. Using the DMA results, the glass transition temperature, storage modulus E’, loss modulus E” and tan δ were measured as a function of temperature, T. The stiffness of the specimens and the temperature range in which the specimens can be subjected to external forces were determined.

## 3. Results with Discussion

### 3.1. Structural Morphology of Samples

#### 3.1.1. Morphology of filaments

From Figure 1, it can be seen that the round shaped neat PLA has a dense structure and uniform transverse surface. 

The round shaped sample PLA-Woodfill_f (Figure 2a) has a rough and highly porous structure with cavities, which may contribute to water transport, as suggested by Le Duigou et al. [15]. As suggested by Tao et al. [20], the interfacial adhesion between wood fibres and the PLA matrix is poor as wood fibres theoretically have a polar (hydrophilic) and PLA a nonpolar (hydrophobic) surface. The weak interfacial adhesion can also be seen in Figure 2b. Wood fibres with clear surfaces are pulled out of the matrix (arrows in Figure 2b), leaving gaps between the fibre and the matrix (dotted circle in Figure 2b) [21]. The PLA-Woodfill_f filament has a very non-uniform transverse surface and structure. 

The sample PLA-Entwined_f has a much denser and non-porous structure compared to the sample PLA-Woodfill_f (Figure 3a). The cavities are smaller, and the fibres are better embedded in the matrix (Figure 3b). Pulled-out hemp fibres were not observed, although gaps between the fibres and matrix could indicate poor interfacial adhesion between the PLA matrix and hemp fibres (Figure 3b). 

#### 3.1.2. Morphology of 3D Printed Samples

Figure 4, Figure 5 and Figure 6 show the fractured surface of the 3D printed samples.

Figure 4a shows the fractured surface of the PLA_3D sample. As it can be seen from the captured image, printed filaments still have a dense structure and are evenly distributed. The boundaries between the printed filaments and layers are clearly visible. While the printed layers are firmly bonded together, both fusion and empty spaces (Figure 4b) are observed between the printed filaments, which may affect the mechanical properties of the sample. 

Figure 5 shows the fractured surface of the PLA_3D sample. As can be seen from the image, the printed filaments are uneven. The structure is different than in the case of the filament. It is no longer as porous as the filament, larger areas of the PLA matrix are observed in the fractured surface within which the wood fibres are better enclosed. However, large empty gaps between the matrix and wood fibres are still present and cavities are visible not only in the fractured surface, but also on the surface along the filament (Figure 5b). Due to the cavities in the structure, the printed filaments are poorly interconnected to each other; however, a weak connection is also visible between the layers. The structure would have been more even if the printing condition had been optimised (as mentioned above, the printing conditions were the same for all three samples). When printing with PLA-Woodfill_f, problems with uneven extrusion occurred. 

Figure 6a shows the fractured surface of the PLA-Entwined_3D sample. The layers are unevenly wide and slightly blurred boundaries between them are still visible. As can be seen from Figure 6b, the printed filaments and layers fused to each other, forming a compact structure with firmly embedded fibres. Between the printed filaments, small triangular-shaped empty spaces are visible, indicating their sequence. 

### 3.2. Colour Differences between Filaments and 3D Printed Samples

Figure 7 shows the filament and 3D printed samples. While PLA is typically “transparent”, its colour changes with the addition of wood and hemp fibres. The dark brown colour of the sample PLA-Entwined_3D, which deepens with the addition of hemp fibres, is, according to Debb and Jafferson [22], a consequence of the oxidation of carbohydrates under high shear stresses and high temperatures. The sample PLA-Woodfill_3D has light brown colour. Usually wood-based filaments are a composite that combines a PLA base material with wood dust, cork and other powdered wood derivatives. The final colour of the filaments depends on the origin of the wood. As mentioned before, Woodfill filaments have the property of changing colour at different extrusion die temperatures, i.e., they become darker at higher temperatures [13].

Table 1 shows that colour values change after the printing when composite filaments are used. In the case of pure PLA-based filament, the colour values hardly changed, while the changes in both filaments with the addition of wood fibres (PLA-Woodfill_3D) and filaments with the addition of hemp fibres (PLA-Entwined_3D) are visible also to the naked eye. The extrusion temperature of 195 °C causes the biocomposite filament to change colour. The results in the literature [14] indicate that the FDM printing process, using different extrusion temperatures, has a substantial impact on the surface colour, density and mechanical properties of the printed wood fibre-reinforced poly(lactic acid) (PLA) composites. The results revealed that most of the physical properties (moisture content, surface roughness, water absorption rate and thickness swelling rate) of the printed samples were not significantly influenced by extrusion temperature, while its density and colour difference increased as the extrusion temperature increased.

### 3.3. Colour Fastness of 3D Printed Samples against Various Agents

Figure 8, Figure 9 and Figure 10 show the average values of the colour differences of the samples printed from commercial filaments exposed to water, oil and detergent.

#### 3.3.1. Colour Fastness of 3D Printed Samples to Water

Following the procedure described in Section 2.2, the samples were exposed to water for 24 h, after which the weight and spectroscopic values of the samples were measured.

The samples were found to have different water uptakes. The largest amount (0.82%) was absorbed by the PLA-Woodfill_3D sample, a much smaller amount (0.38%) by the PLA_3D sample, while no water absorption was observed for the PLA-Entwined_3D sample. These results were expected. Although the PLA matrix is hydrophobic, the PLA_3D sample has empty spaces in the structure (Figure 4) as well as pores on the surface, which could contribute to the slight capillary water absorption. The PLA-Woodfill_3D sample has a higher proportion of hydrophilic wood fibres in the PLA matrix that absorb water more readily. Moreover, it has a lot of visible voids and cavities (Figure 2 and Figure 5), which contribute to higher water penetration into the sample. The PLA-Entwined_3D sample had a low percentage of hydrophilic hemp fibres and a rather compact structure without visible voids or pores on the surface (Figure 6), resulting in a less absorbent sample. 

It can be seen from Figure 8 that the colour of the PLA and PLA-Entwined samples changes slightly on contact with water, but less than the PLA-Woodfill samples, the differences still being small. The average differences range from 0.53 to 1.82, which means that most of the colour differences are not perceptible to the eye. 

#### 3.3.2. Colour Fastness of 3D Printed Samples to Oil

A similar tendency as for water absorption was also found for oil sorption, which was expected, since PLA is not only a hydrophobic but also oleophilic polymer [23]. Thus, the structural morphology with empty spaces, voids and cavities influences the oil sorption [24]. Accordingly, the PLA-Entwined_3D sample with a compact structure absorbed the smallest amount of oil after 48 h (0.32%), the PLA_3D sample with empty spaces and smaller voids absorbed slightly more oil (0.61%), while in the case of the PLA-Woodfill_3D sample, a significantly higher amount of oil was absorbed by the voids and cavities, and some of it may remain in the structure (3.59%). 

From Figure 9, it can be seen that there were large colour differences in all samples exposed to oil. The largest differences were seen in PLA-Entwined, and the smallest differences for the PLA and PLA-Woodfill samples. All of the colour differences listed are above 2 and can therefore be seen with the naked eye.

#### 3.3.3. Colour Fastness of 3D Printed Samples to Detergent

The detergent absorption after three hours was the highest again for PLA-Woodfill_3D (0.75%), smaller for PLA_3D (0.23%), while the sample PLA-Entwined_3D did not absorb detergent.

The samples made from commercially available filaments are generally colour resistant to detergents (Figure 10) as colour differences are not visible to the naked eye.

#### 3.3.4. Colour Fastness of 3D Printed Samples to Light

Table 2 shows the L*, a* and b* values of the 3D printed samples, measured before and after the exposure to Xe light, and the calculated colour differences ΔE*ab.

From the calculated colour differences, the sample PLA-Woodfill_3D is the least resistant to light. The colour difference before and after the exposure is obvious to the naked eye (darkened part of the sample in Figure 8), and the measured colour difference is very high, i.e., over 6. The lightness values (L*) did not change significantly, nor did the value a* change significantly. The differences occurred mainly in the value b* (colour saturation in the yellow spectrum), which increased by about 6; hence, the large colour difference. PLA_3D and PLA-Entwined_3D were more lightfast, and their measured colour differences ΔE*ab ranged from 1 to 3. For the printed samples made of the PLA filament, PLA_3D, the differences were very small, differing most at the b* values, which decreased to almost 0. For the PLA-Entwined_3D sample, the values of L*, a* and b* increased, with smaller measured differences. A higher L* value is also visible to the naked eye (Figure 11). In the research by Mikołajczyk and Kuberski [25], it was proved that UV light significantly changed the optical properties of PLA.

#### 3.3.5. Colour Fastness of 3D Printed Samples to Temperature 

Table 3 shows the spectrophotometric values before and after the exposure of samples to higher temperatures and the calculated average values of colour differences. For the samples exposed to 80 °C, small colour differences, between the values close to 3 and 4, were observed for all samples. For the samples PLA_3D, the value L* increased slightly, and slightly larger differences are seen in the value b*, which dropped by almost 2, indicating a decrease in colour saturation. The differences in saturation and brightness were greater for PLA-Woodfill_3D. The PLA-Entwined_3D samples showed that exposure at 80 °C had the greatest effect on lightness, which increased by almost 4, while the saturation values remained similar.

For the samples exposed to 110 °C, the PLA-Woodfill_3D samples stood out the most, where the average colour difference reached 11. It was affected in all three values of lightness and saturation. Lightness decreased and saturation increased, which can be seen with the naked eye as a visible darkening of the samples. For the PLA_3D samples, the colour differences were similar to those at 80 °C, and for the PLA-Entwined_3D samples, the colour differences increased by 1 on average, from 4 to 5 (Table 3). This time, the temperature affected not only the lightness of the samples but also the saturation (especially for b*).

The colour differences increased to 7 for the PLA_3D samples, lightness (L* value) decreased by 5, and the saturation value b* increased by 4, which was reflected in a slightly yellowish coloration.

Temperature had the smallest effect on the PLA-Entwined_3D samples as the average colour difference was less than 4 (Table 3). The changes were mainly observed in the saturation values a* and b*, less in lightness L*.

The samples exposed to a temperature of 130 °C showed the largest differences in the PLA-Woodfill_3D samples, where the average colour difference was up to 26. Deformations in the shape and large colour changes were also easily visible to the naked eye as the light brown became a burnt, dark colour. The value L* decreased by about 25, and a large difference was also caused by the value a*, which increased by 10. Both confirm the dark and saturated colour of the samples.

The results show that with temperature changes, the colour differences are the smallest for the PLA-Entwined_3D samples and the largest colour differences occur for the samples made with wood fibre filaments, PLA-Woodfill_3D, which have very low resistance to higher temperatures.

### 3.4. Dynamic Mechanical Analysis (DMA) of Filaments and 3D Samples

DMA was carried out to investigate the effects of hemp and wood fibre loading on the viscoelastic properties of PLA composites. The viscoelastic curves of the bending storage modulus, E’, of filaments and 3D printed samples are shown in Figure 12. All the samples exhibited a significant fall in the storage modulus in the regions between 42 and 70 °C, and 60 and 77 °C for the filaments and 3D printed samples, respectively, indicating the transition from glassy to rubbery state of the materials. The α-temperature relaxation transition happened in a very narrow temperature region, indicating the dominant amorphous molecular structure of all samples. 

The **storage modulus** is an indicator of the stiffness of a material. In the glassy region (at 40 °C), the highest value of the storage modulus was identified for the PLA-Entwined_f sample (6.8 GPa), slightly lower for the PLA_f (5.8 GPa) and the lowest for the PLA-Woodfill_f sample (2.5 GPa) (Table 4). In general, natural fibres act as a rigid, reinforced filler that increases stiffness [21,26,27], but only in the case when the content of the added fibres is low. With increasing fibre content, the stiffness starts to drop [28,29]. The content of hemp fibres in the sample PLA-Entwined_f was relatively low, which resulted in a higher value of the storage modulus. In contrast, the sample PLA-Woodfill_f had a much higher content of wood fibres and thus a lower storage modulus. 

However, some additional factors may influence the storage modulus as reported in some studies. In the studies by Sawpan et al. [29] and Cristea et al. [30], it was reported that added hemp fibres increase the formation of crystallinity due to the increased availability of nucleation sites and thus increase the storage modulus of the material. According to Cristea et al. [30] and Battegazzore et al. [28], a lower content of hemp fibres contributes to better dispersion in the PLA matrix, resulting in a more efficient stress transfer, while at higher loading, the filler starts to aggregate, acting as vulnerable points in the composites, or the PLA matrix can space out the PLA chains leading to a lack of interaction between the fibres and PLA matrix. The reinforcement effect of added hemp in the PLA matrix is also reported in numerous other studies [31,32,33,34]. Nevertheless, it is also important to add that the lower storage modulus in the PLA-Woodfill_f sample is probably also connected with the significantly higher porosity of the sample (Figure 2). This porosity phenomenon leads to an inhomogeneous morphological structure and deterioration of the PLA matrix. The formation of empty voids hinders the interaction between the fibres and the matrix, leading to a lower storage modulus.

As shown in Figure 12, the **storage modulus of all 3D printed samples** dropped in the glassy region (40 °C). The highest decrease is indicated for the PLA-Woodfill_3D sample and the smallest for the sample PLA-Entwined_3D. The higher content of wood fibres caused an inhomogeneous morphological surface structure, and the correlation between the modulus and the surface morphological structure was again confirmed. The samples with a uniform structure of printed layers, where the layers fit well together (PLA_3D and PLA-Entwined_3D, Figure 4a and Figure 6a), showed higher storage modulus E’ values (2.7 GPa and 2.5 GPa for PLA_3D and PLA-Entwined_3D, respectively). In the process of 3D printing with fused deposition modelling (FDM), the material (filament) in the heating nozzles melts to a partially liquid state, in which polymer chains are rearranged in both amorphous and crystalline regions and are oriented in the direction of extrusion/flow. Rapid cooling after the transition of the polymer liquid from the nozzle and loading into the layer “freezes” the newly rearranged and oriented polymer chains. Such a rearrangement of macromolecular chains, in addition to the conditions in 3D printing, affects the final characteristics of the samples. If the extrusion of polymer fluids from the nozzle is disturbed for any reason (e.g., nozzle clogging due to additives, non-optimised printing conditions etc.), there may be a drop in the modulus due to a reduction in adhesion between the structural components in the material, i.e., the porosity of the surface seen when wood fibres are added. A large increase in porosity is also a result of a higher decrease in the viscosity of the PLA melt flow. Consequently, decreased melt flow from the nozzle results in very poor adhesion between the layers in the printed specimen [22]. Due to poor interfacial adhesion between the layers, the stiffness decreased. The result is thus in accordance with the morphological results of the 3D printed samples seen in the SEM captured images (Figure 5).

The deterioration of the stiffness of the 3D printed sample with wood fibres (PLA-Woodfill_3D) is also in correlation with the poorer mechanical properties of the 3D samples (unexposed samples), indicating the poor stress transfer from the PLA matrix to the wood fibres.

**Damping** defines the way in which a material absorbs and disperses energy and is a good provider of information about the internal friction of the material and the adhesion of the interface between the matrix and filler. As stated by Baghaei et al. [34], the molecular motion in the interphase of a composite will contribute to damping. In Figure 13, the damping behaviour (tan δ vs. temperature) of the filaments and 3D printed samples is presented, while Figure 14 demonstrates the loss modulus of the samples, i.e., the ability of the samples to dissipate the heat rather than store it (viscous properties).

**Tan δ in the case of filaments** is the highest for the PLA-Entwined_f sample (2.1), lower for the PLA_f sample (1.1) and the lowest for the PLA-Woodfill_f sample (1.0) (Figure 13). The presence of a small content of hemp fibres increased the segmental motions of the PLA filament matrix during the glass transition of the filament and the structure would rather dissipate (the load in the form of scattered heat) than store it. The material behaves more viscously than elastically. The same trend is observed in the peaks of loss modulus (Figure 14), which represents the viscous component of the samples. In the case of the PLA-Woodfill_f sample, the segmental motion is hindered due to a higher proportion of added fibres, which gave the structure greater potential to store the applied external load rather than dissipating it. The material behaves more elastically than viscously. 

In the case of the **3D printed samples**, the highest tan δ and loss modulus peaks are indicated for the neat PLA_3D sample (1.7), slightly lower for the PLA-Entwined_3D (1.5) and the lowest for PLA-Woodfill_3D sample (0.8) (Table 4). During the 3D printing, the molten PLA component with higher viscosity surrounds the fibres tighter, resulting in greater adhesion between the fibres and matrix (Figure 5b and Figure 6b). These better embedded hemp and wood fibres decreased the segmental motion of the PLA matrix compared to the filament samples. 

The lowest damping of the PLA-Woodfill_f and PLA-Woodfill_3D samples can also be attributed to a significantly higher portion of wood fibres. The damping of polymers is much greater than the damping of natural fibres [35]. 

Tan δ peak position also defines the **glass transition temperature**, which is presented in Table 4. The glass transition temperature (Tg) of all filaments is significantly lower than that of 3D samples. The following results are thus observed:-Tg of the neat PLA_f was 59.4 °C which increased to 75.2 °C for PLA_3D due to the formation of a dense 3D structure with printed layers firmly bonded together (Figure 4a) and, most importantly, due to a more substantial fraction of rigidly confined PLA macromolecules, which was also reported by Pop et al. [36]. They stated that an increase in this fraction in the 3D printing process causes an increase in glass transition temperatures, despite the decrease in the overall crystallinity of the 3D printed object. The latter indicated that the PLA structure does not have enough time to crystallise in a more ordered manner, given the rapid cooling gradient of the molten extruded material.-When compared to the neat PLA, the decreased Tg in PLA-Woodfill_f (56.1 °C) and PLA-Woodfill_3D (69.3 °C) is probably a consequence of poorer interaction between the fibres and PLA matrix (which is also in correlation with a lower storage modulus; Figure 2 and Figure 5) and of the high porosity of the structure.-When 3D printed, the Tg of PLA-Woodfill_3D increased when compared to PLA-Woodfill_f. This increase is probably due to the significant porosity of its structure and inversely related to the thermal conductivity which increases linearly with bulk density of the structure: materials with higher porosity will take a longer time to reach the desired temperature, resulting in higher detected glass transition temperatures [37]. Additionally, during the DMA testing, frictional forces arise in the structure due to the constant supplied oscillatory strain. As suggested by Ross et al. [37], more friction arises in a sample with lower porosity due to the closer proximity of the structure on a macroscopic level. This may serve as an additional energy source to the thermal energy being supplied to the material during the DMA testing. Hence, more energy is associated with less-porous samples than more-porous samples at the same temperature. This might also serve as the explanation for the more-porous PLA-Woodfill_3D sample showing a higher glass transition temperature.-The highest increase was detected in the samples with hemp fibres, namely, the Tg of the PLA-Entwined_f sample was 51.8 °C which increased to 77.3 °C for the sample PLA-Entwined_3D. This was again a result of the formation of a compact 3D structure with firmly embedded fibres (Figure 6a).

Due to the low glass transition temperatures of the PLA structure, resulting in a rapid decrease in the storage modulus at low temperatures (softening), the filaments could be safely used in temperatures of up to 50, 40 and 45 °C for PLA_f, PLA-Woodfill_f and PLA-Entwined_f, respectively. The 3D printed samples can be used at higher temperatures of up to 63, 65 and 60 °C for PLA_3D, PLA-Woodfill_3D and PLA-Entwined_3D, respectively. Additionally, since the mechanical properties may decrease drastically above the glass transition temperature [38], the temperature region where the handling of the final products should be considered with great caution and with a limited external stress is between 55 and 80 °C (the area below the tan δ peak; Figure 13). 

The determination of the safe temperature region for the final use of 3D samples by DMA also resulted in the mechanical testing of some basic mechanical parameters (breaking force, elongation and Young’s modulus) of 3D samples at room temperature. The results presented in Table 5 show that the samples with the addition of hemp and wood fibres had different mechanical properties. The force required for the PLA-Woodfill_3D sample to break was significantly lower (by 32.3%), and that for the PLA-Entwined_3D sample was significantly higher (by 53.4%), than for the neat PLA_3D. The PLA-Woodfill_3D sample was also less extensible, (elongation at break was by 22.4% lower) and PLA-Entwined_3D more extensible (elongation at break was by 18.9% higher) than PLA_3D. According to the SEM images, the samples have a very different structure, which influenced the mechanical properties. The uneven layered structure of the 3D printed samples (especially PLA_3D and Woodfill_3D), confirmed with SEM images, leads to the conclusion that the optimisation of the 3D printing conditions is needed to gain more relevant and reliable results of the mechanical properties.

## 4. Conclusions

The aim of the research was to determine the colour fastness of 3D printed samples that can be used as decorative and/or useful household items. These are items such as household utensils, jewellery, clothing accessories, unique art products and toys. Nowadays, with increasing environmental awareness, more and more decorative and useful items are being produced with 3D printing using biocomposite materials. With the technology of extruding biocomposite filaments containing various additives, beautiful objects can be made; however, their durability remains questionable.

In decorative and/or useful objects, where the scope of application of biocomposite materials is broad, both mechanical properties and colour fastness are important, affecting not only the appearance, but also the satisfaction and quality of user experience. The relationships between the colour values and colour differences at different stages of the product use cycle, and the mechanical and chemical properties are important as well. This is particularly true for household products, toys and fashion accessories. Such products come into contact not only with light, moisture, dust and dirt, but also with water, detergents, oils, creams and saliva.

The aim of the study was achieved. We found that the additives used in polymer matrix, in our case, added wood and hemp fibres and residues, affect the colour fastness. In particular, the colour fastness to light and temperature was reduced compared to pure PLA. This effect was pronounced in the PLA-Woodfill samples. 

We found that the colour values of printed samples with composite filaments changed compared to the original filaments during the printing process. 

The analysis of the captured images on SEM showed that the biocomposite filament with a higher proportion of wood fibres had the most porous structure, while the sample made of pure PLA and the sample with a lower proportion of hemp fibres were significantly more compact, although some voids were found in the latter. After the 3D printing, the samples again had different and distinct morphological structures, which also influenced the properties of each sample. The morphology of the pure PLA_3D had a fairly regular structure, but some voids were observed between the filaments and layers. The 3D sample with hemp had the most compact morphology, with the filaments and layers generally fused to each other. The most irregular structure was observed in the sample with wood fibres, which had voids and cavities in the filaments themselves, between them and between the layers, which drastically affected the mechanical and dynamic mechanical properties. The structure of the 3D samples could be improved by optimising the conditions during printing. However, the aim of the research was not to obtain the most optimal structure of the 3D printed samples; hence, factors in the printing were not studied.

All printed samples showed very good resistance to detergent, less to water and the worst to oil. 

The samples were found to have different water and oil uptake. The largest percentage of water was absorbed by the PLA with added wood fibres, while no water absorption was observed for the printed samples PLA with added hemp fibres. 

A similar tendency as for water absorption was also found for oil sorption, which was expected. The 3D printed sample with added hemp fibres with a compact structure absorbed the least amount of oil. In the case of the printed PLA with added wood fibres, a significantly higher amount of oil was absorbed by the voids and cavities, and some of it may remain in the structure.

The worst colour fastness after the Xenotest, i.e., irradiation with Xe light, was shown by the sample with wood fibres, followed by the sample with hemp fibres, whereas the pure PLA proved to have the best colour fastness. After the 3D printed samples were exposed to high temperatures, the highest colour degradation occurred in the samples printed from the composite filament with wood fibres.

The DMA results revealed that adding hemp fibres showed the reinforcement effect in the filament, probably due to the content of hemp fibres being sufficiently low, compared to the content of wood fibres, which resulted in better dispersion in the PLA matrix and lower porosity of the PLA matrix structure. In contrast, the highly porous and thus inhomogeneous morphology of the PLA matrix with added wood fibres reduced the stiffness of the filament. The printing process additionally had an enormous effect on the elasticity and glass transition of the PLA material, with or without added fibres. The stiffness of all printed samples was significantly reduced and was the lowest when wood fibres are included. A highly porous structure is additionally altered with the formation of large empty voids and poorly fitted printed layers. The glass transition temperature of all printed samples was increased when compared to the filaments, probably due to the formation of rigidly confined PLA macromolecules fractions. Since softening is the main weakness of the PLA material (a consequence of the low glass transition of the PLA polymer), additional caution should be used when choosing the appropriate temperature region for the use of printed samples. Safe use, without the occurrence of the softening of the material, is thus limited to 63, 65 and 60 °C for PLA_3D, PLA-Woodfill_3D and PLA-Entwined_3D, respectively. Additionally, to avoid the formation of “microcracks” in the material, which appeared due to increased external stress, the recommended temperature region is defined as between 55 and 80 °C. 

The mechanical testing showed that the samples with the addition of hemp and wood fibres exhibited different mechanical properties. The force required to break was significantly lower for the printed PLA samples with added wood, and significantly higher for the printed PLA samples with added hemp, compared to the neat PLA samples. The printed sample with wood fibres was also less extensible, while the printed samples based on PLA and PLA with hemp fibres were more extensible. For further research, it will be necessary to optimise the printing conditions separately for each of the filaments used. This will allow a more precise analysis of the comparison between the mechanical properties of each sample.

Based on the research, it can be concluded that the samples based on PLA with the addition of wood fibres were the most sensitive to various factors. Due to the open porous structure, they were sensitive to both water and oil. When exposed to light, their colour changed significantly. As the temperature increased, the elasticity and the mechanical properties of these samples deteriorated drastically. Through our research, we have proven that it is necessary to pay attention to the type of biocomposite materials used when making decorative and/or useful household items.

## Figures and Tables

**Figure 1 polymers-13-03738-f001:**
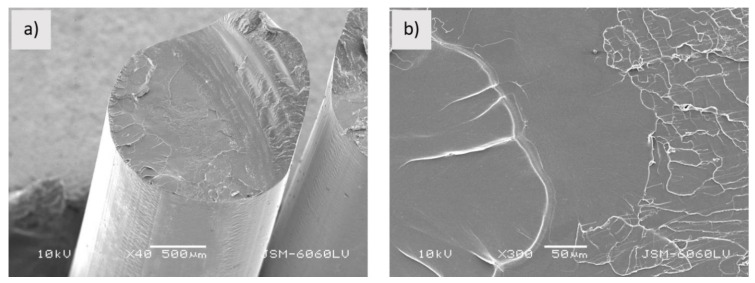
Sample PLA_f; (**a**,**b**) fractured surface (mag. (**a**) 40×, (**b**) 300×).

**Figure 2 polymers-13-03738-f002:**
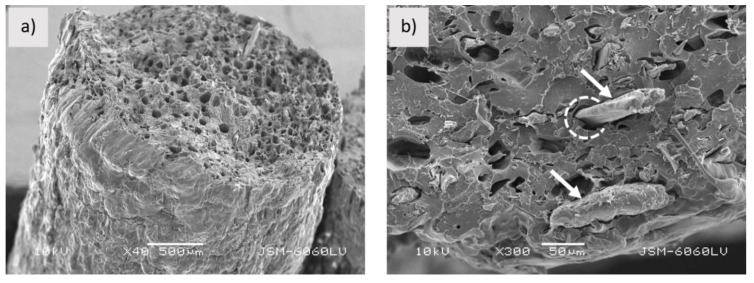
Sample PLA-Woodfill_f; (**a**) fractured surface (mag. 40×), (**b**) pulled out fibres (pointed by arrows) and gaps between PLA matrix and wood fibres (indicated by dotted circle) (mag. 300×).

**Figure 3 polymers-13-03738-f003:**
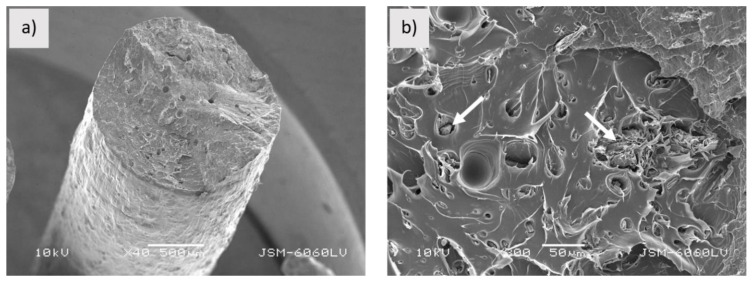
Sample PLA-Entwined_f; (**a**) fractured surface (mag. 40×), (**b**) gaps between matrix and hemp fibres (pointed by arrows) (mag. 1200×).

**Figure 4 polymers-13-03738-f004:**
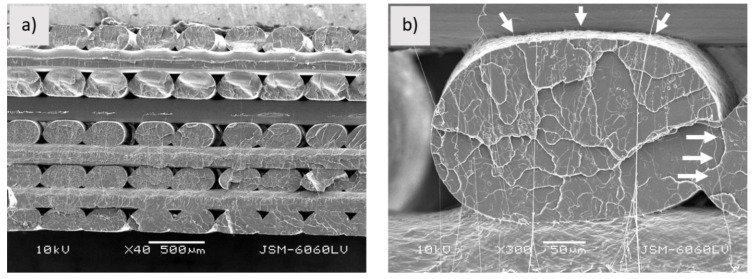
Sample PLA_3D; (**a**) fractured surface (mag. 40×), (**b**) printed filaments (area where two printed filaments and layers are firmly connected are pointed by arrows) (mag. 300×).

**Figure 5 polymers-13-03738-f005:**
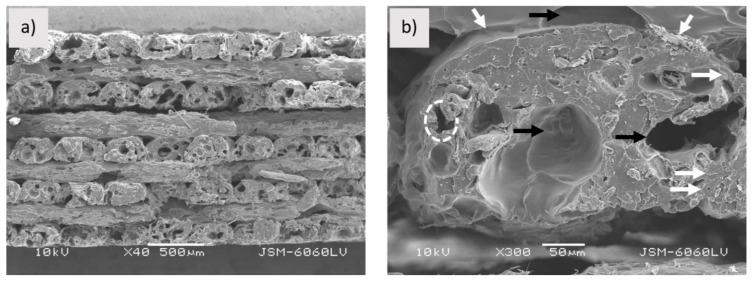
Sample PLA-Woodfill_3D; (**a**) fractured surface of sample (mag. 40×), (**b**) fractured surface of printed filament (white arrows point to area where two printed filaments and layers are firmly connected; dotted circle shows gap between wood fibre and matrix, black arrows point to cavities) (mag. 300×).

**Figure 6 polymers-13-03738-f006:**
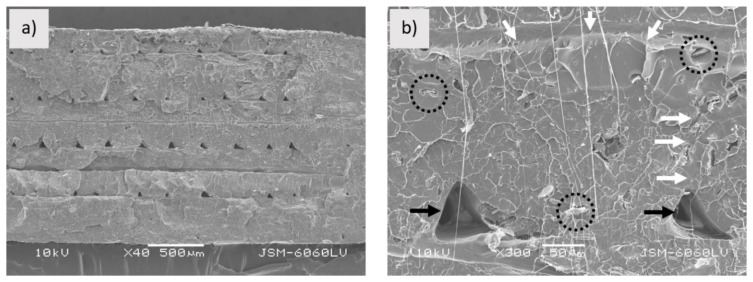
Sample PLA-Entwined_3D; (**a**) fractured surface of sample (mag. 40×), (**b**) fractured surface of print filament (white arrows indicate area where two printed filaments and layers are firmly connected; black arrows point to empty space; dotted circles show embedded hemp fibres) (mag. 300×).

**Figure 7 polymers-13-03738-f007:**
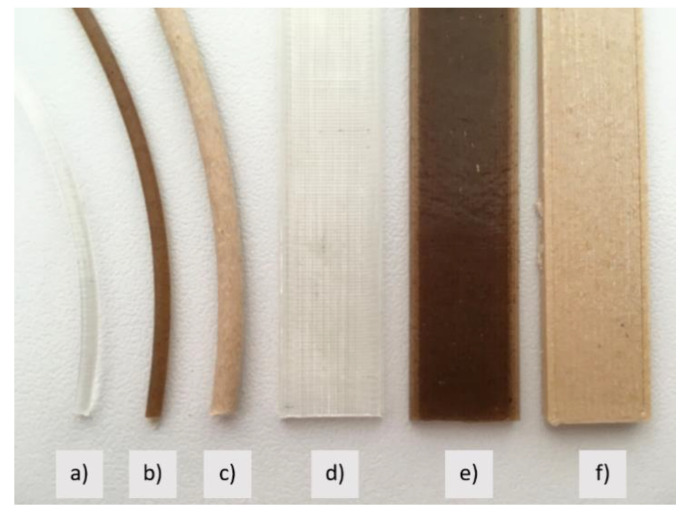
Samples of filaments: (**a**) PLA_f, (**b**) PLA-Entwined_f, (**c**) PLA-Woodfill_f, and 3D prints: (**d**) PLA_3D, (**e**) PLA-Entwined_3D and (**f**) PLA-Woodfill_3D.

**Figure 8 polymers-13-03738-f008:**
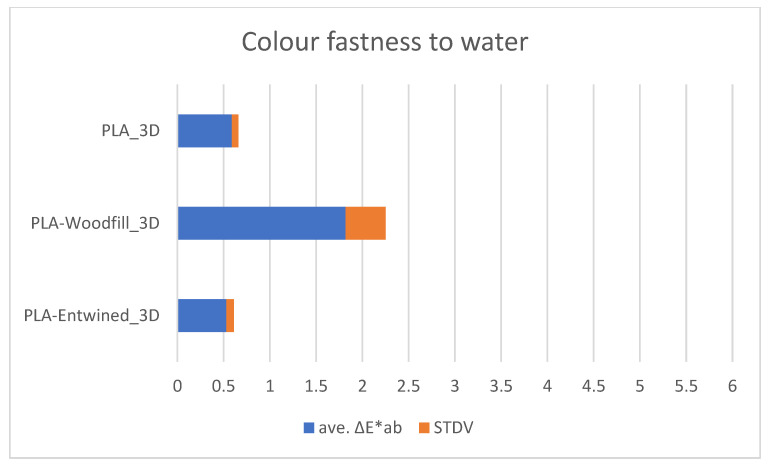
Colour fastness to water.

**Figure 9 polymers-13-03738-f009:**
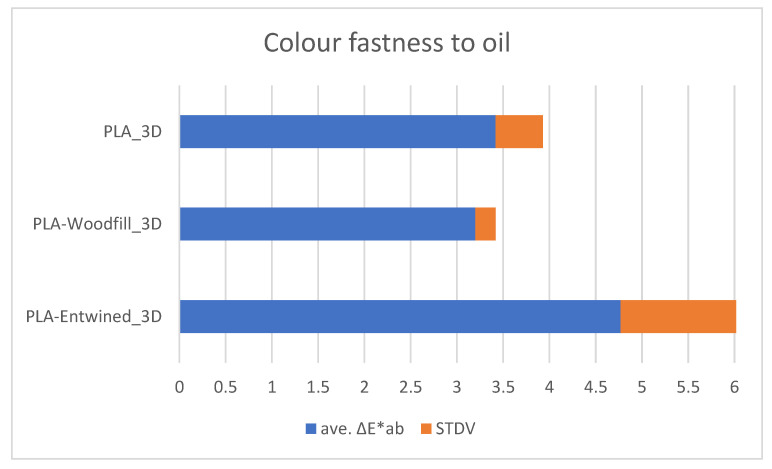
Colour fastness to oil.

**Figure 10 polymers-13-03738-f010:**
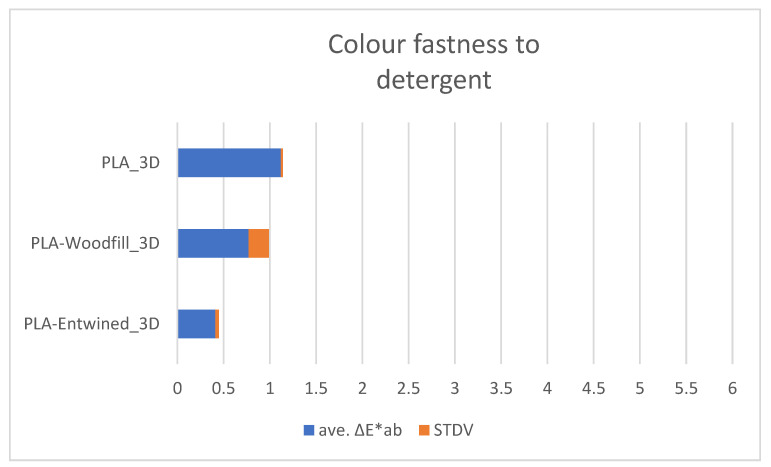
Colour fastness to detergent.

**Figure 11 polymers-13-03738-f011:**
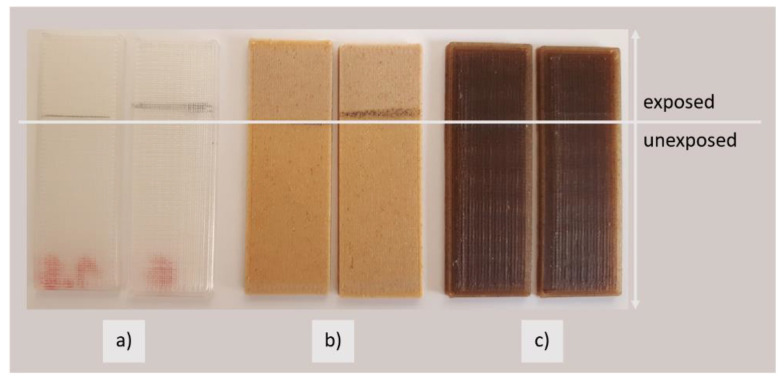
Samples after exposure to Xe light; bottom: unexposed part, top: part exposed to Xe light; (**a**) PLA_3D, (**b**) PLA-Woodfill_3D, and (**c**) PLA- Entwined_3D.

**Figure 12 polymers-13-03738-f012:**
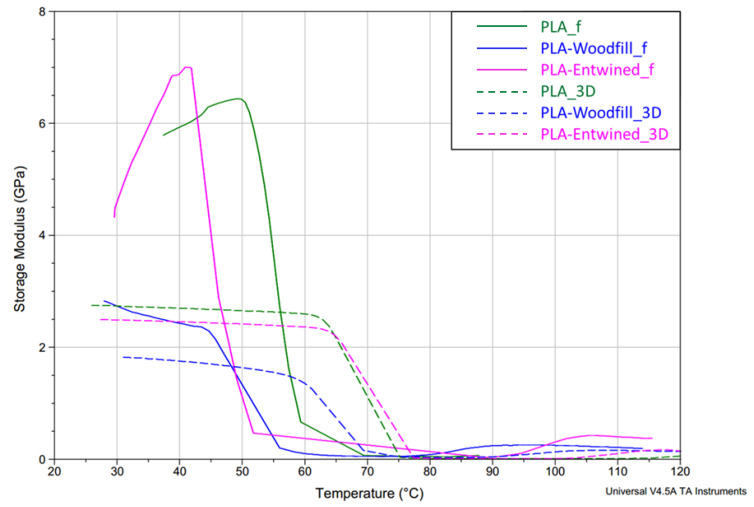
Storage modulus of samples PLA, PLA-Woodfill and PLA-Entwined before and after 3D printing vs. temperature at 10 Hz.

**Figure 13 polymers-13-03738-f013:**
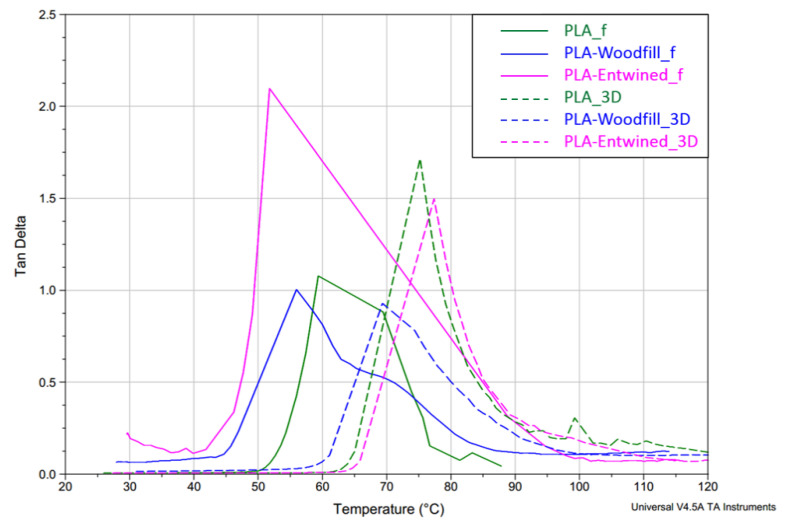
Tangent delta of PLA, PLA-Woodfill and PLA-Entwined samples before and after 3D printing vs. temperature at 10 Hz.

**Figure 14 polymers-13-03738-f014:**
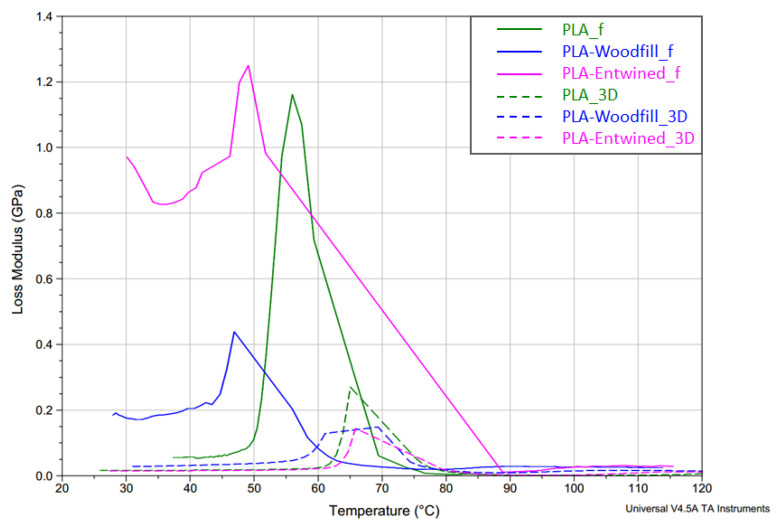
Loss modulus of PLA, PLA-Woodfill and PLA-Entwined samples before and after 3D printing vs. temperature at 10 Hz.

**Table 1 polymers-13-03738-t001:** Colour values of filaments and 3D printed samples.

Sample	Filament (f)			3D Printed Sample (3D)			ΔEab*
	L*	a*	b*	L*	a*	b*	
PLA	55.49 ± 1.15	–0.26 ± 0.2	2.17 ± 0.32	57.05 ± 0.21	–0.15 ± 0.14	1.47 ± 0.35	1.71 ± 0.81
PLA-Woodfill	65.95 ± 3.32	4.99 ± 0.23	18.06 ± 0.52	68.24 ± 0.28	5.70 ± 0.13	21.81 ± 0.31	4.45 ± 1.64
PLA-Entwined	30.33 ± 2.16	4.67 ± 0.24	7.76 ± 0.36	33.12 ± 0.4	5.57 ± 0.44	13.10 ± 1.17	6.09 ± 1.63

**Table 2 polymers-13-03738-t002:** L*, a*, b* and ΔE*ab values of 3D printed samples before and after exposure to light; mean value ± SD.

Sample	Before Exposure to Light			After Exposure to Light			
	L*	a*	b*	L*	a*	b*	∆E*ab
PLA_3D	51.79 ± 0.01	–0.01 ± 0.03	1.59 ± 0.06	51.63 ± 0.35	0.15 ± 0.09	0.28 ± 0.32	1.36 ± 0.33
PLA-Woodfill_3D	68.38 ± 0.10	5.24 ± 0.06	21.55 ± 0.05	68.99 ± 0.14	6.24 ± 0.05	27.82 ± 0.04	6.39 ± 0.01
PLA-Entwined_3D	28.66 ± 0.53	4.84 ± 0.23	8.54 ± 0.30	30.54 ± 0.45	5.38 ± 0.23	9.3 ± 0.37	2.11 ± 1.23

**Table 3 polymers-13-03738-t003:** L*, a*, b* and ΔE*ab values of 3D printed samples before and after exposed to high temperatures (80 °C, 110 °C, 130 °C); mean value ± SD.

Sample	T [°C]	Before Exposure to Temperature			After Exposure to Temperature			∆E*ab
		L*	a*	b*	L*	a*	b*	
PLA_3D	80	53.73 ± 0.65	–0.50 ± 0.01	2.87 ± 0.22	55.3 ± 0.63	–0,64 ± 0.28	0.50 ± 0.34	2.94 ± 0.49
	110	56.23 ± 1.19	0.07 ± 0.05	1.19 ± 0.09	54.25 ± 0.53	–0.87 ± 0.03	2.68 ± 0.46	2.79 ± 1.00
	130	56.50 ± 0.83	–0.35 ± 0.02	2.46 ± 0.15	51.03 ± 1.07	–0.55 ± 0.06	6.76 ± 0.61	7.08 ± 0.59
PLA-Woodfill_3D	80	69.04 ± 0.11	5.32 ± 0.07	21.66 ± 0.27	68.05 ± 0.30	7.24 ± 0.15	23.42 ± 0.31	2.81 ± 0.30
	110	69.42 ± 0.22	5.60 ± 0.13	21.9 ± 0.28	61.49 ± 1.74	10.77 ± 1.09	27.80 ± 0.66	11.17 ± 2.03
	130	69.31 ± 0.33	5.47 ± 0.11	21.25 ± 0.60	44.27 ± 3.51	15.26 ± 0.04	23.24 ± 1.77	26.96 ± 2.92
PLA-Entwined_3D	80	30.11 ± 1.34	4.80 ± 0.26	9.33 ± 1.16	33.85 ± 0.78	4.70 ± 0.18	9.82 ± 0.57	3.81 ± 0.64
	110	31.02 ± 0.91	4.75 ± 0.23	8.65 ± 1.10	34.72 ± 0.56	5.80 ± 0.34	11.73 ± 0.93	4.94 ± 0.57
	130	32.44 ± 1.32	4.8 ± 0.26	9.53 ± 1.32	30.86 ± 0.92	7.19 ± 0.44	11.44 ± 1.02	3.6 ± 0.18

**Table 4 polymers-13-03738-t004:** Glass transition temperature (Tg), bending storage modulus (E’) at 40 °C, damping factor (tan δ) and temperature region sensitive to high mechanical loading of samples.

	Tg [°C]	E’ at 40 °C [GPa]	tan δ Peak [/]	Temperature Region Sensitive to High Mechanical Loading [°C]
PLA_f	59.4	5.8	1.1	50–75
PLA-Woodfill_f	56.1	2.5	1.0	45–85
PLA-Entwined_f	51.8	6.8	2.1	45–90
PLA_3D	75.2	2.7	1.7	65–80
PLA-Woodfill_3D	69.3	1.8	0.8	55–75
PLA-Entwined_3D	77.3	2.5	1.5	65–80

**Table 5 polymers-13-03738-t005:** Mechanical properties of 3D printed samples at room temperature; mean value ± SD.

Sample	Breaking Force [N]	Elongation [%]	Young’s Modulus [MPa]
PLA_3D	387.28 ± 28.98	5.84 ± 0.50	963.81 ± 97.51
PLA-Woodfill_3D	180.56 ± 18.63	4.53 ± 0.25	523.48 ± 62.38
PLA-Entwined_3D	512.41 ± 21.16	7.20 ± 0.66	1038.11 ± 59.56

## Data Availability

Data are contained within the article.

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
