# Peer review of "Colour Fastness to Various Agents and Dynamic Mechanical Characteristics of Biocomposite Filaments and 3D Printed Samples"

_polymers, 2021, doi:10.3390/polym13213738_

Round 1

Reviewer 1 Report

Please show the loss modulus for each sample to analyze the viscous propertiers.

Author Response

Dear Sirs,

we have reviewed and answered your questions. We hope that the desired quality is now achieved. If anything else needs to be changed, we will be happy to do so. Nice regards, Deja

Reviewer 2 Report

Comments to Author:

This manuscript describes the colour fastness of biocomposites filaments for 3D printed samples. Overall, this is a confusing study because some literature already point put that PLA has an inflammatory reaction result in safety concerns. Inn addition, it requires more details and editing for publication.

(1) In Chapter 2.2 Methods, Formula in Line 212, there’s no explanation of what is L*, a* and b*

(2) Line 213-214, “accordance with the standard SIST ISO 2836:2004 Graphic Technology – Prints and Printing Inks”. This SIST ISO specifies methods of assessing the resistance of printed materials to liquid and solid agents, solvents, varnishes, and acids. How to jump to the conclusion that this standard can be used for solid material. Specifically, 3D printed samples? Any reference pointing about this?

(3) Line 229, almost the same with comments no 2, SIST ISO 12040 is Assessment of light fastness using filtered xenon arc light, how to jump to the conclusion that this standard can be used for solid material?

(4) Line 238, Temperature selected for this research is 80oC, 110oC and 130oC. What’s the baseline for choosing this temperature (any reference journal?). And it’s mentioned in the manuscript that the application for this 3D printed samples is for decorative or household items. Usually, the temperature applied for this application is room’s temperature (around 37oC) that doesn’t relate to 80oC, 110oC and 130oC.

(5) Line 241, What standards are used (ASTM, etc.) for the mechanical properties analysis?

(6) Still related to comments number 5, how to compare the mechanical properties for the filament and the 3D printed samples, primarily associated with the testing sample. How to create this mechanical properties testing sample for the filament was not explained well in the methods.

(7) In Chapter 3.3. COLOUR FASTNESS OF 3D PRINTED SAMPLES TO VARIOUS AGENTS Line 347-394. On this result, the manuscript fails to shows a connection between the figure and the explanation. The figure shows colour fastness, but the description mentions some percentage that did not relate to the figure and was not written in any table or the manuscript.

(8) Also, for Colour fastness results, the writer’s failed to explain why there’s a difference in colour fastness for each material treated with various agents (water, oil, and detergent).

(9) In chapter 4. Conclusions, Line 624-626. The conclusion that added wood and hemp fibres and residues strongly affect the colour fastness and mostly reduce it compared to pure PLA seems didn't match with the research result. Colour fastness to water, wood fill is bad (compared to pure PLA), Colour fastness to Oil, Entwined is bad, and colour fastness to detergent, PLA is the best, but it still under naked eye appearance threshold (under 2). This conclusion needs to be reconstructed.

(10) There's a lot of writing error in the manuscript, such as missing link reference (line 397, line 418), the same picture appears twice (line 316 and line 579) and not to mention, probably spread out of un-consistent font at almost all over manuscript paragraphs, which tend to shows how the seriousness of the writer to their manuscript.

(11) The most important thing is biocompatible test. 

One typical reference paper can be checked :

YuvalRamot, MoranHaim-Zada, Abraham J.Domb, AbrahamNyska. "Biocompatibility and safety of PLA and its copolymers." Advanced Drug Delivery Reviews, 107: 153-162.

Author Response

Dear Sirs!

We have reviewed and answered your questions. We hope that the desired quality is now achieved. If anything else needs to be changed, we will be happy to do so.

Nice regards, Deja

Reviewer 3 Report

A typical experimental study in the field of practical polymer materials science. Nevertheless, I rate the presented results as original and recommend the article for publication. 
When preparing the manuscript for publication, I recommend the authors to compare in the text of the article the lightfastness of traditional fibres and yarns with samples obtained by 3D printing, and, secondly, to explain more thoroughly the reasons of exposure of the samples to oils!!!

Author Response

(The authors gave the same response as above.)
